# Top-Down Neural Model For Formulae

**Karel Chvalovský**
Czech Institute of Informatics, Robotics, and Cybernetics
Czech Technical University in Prague
karel@chvalovsky.cz

## Abstract

We present a simple neural model that given a formula and a property tries to answer the question whether the formula has the given property, for example whether a propositional formula is always true. The structure of the formula is captured by a feedforward neural network recursively built for the given formula in a top-down manner. The results of this network are then processed by two recurrent neural networks. One of the interesting aspects of our model is how propositional atoms are treated. For example, the model is insensitive to their names, it only matters whether they are the same or distinct.

## 1 Introduction

In real-world situations a very successful approach, popularized in Kahneman (2011), to problem solving is based on a clever combination of fast instinctive (heuristic) reasoning and slow logical reasoning. The latter is exemplified by abstract logical formulae where only structural properties matter. If computers are involved, a logical formula is traditionally a syntactic object which is a subject to simple but very fast syntactic manipulations Robinson & Voronkov (2001). Hence all but very basic decisions are postponed if possible. However, this viewpoint is rapidly changing as various AI methods are tested in the field of automated reasoning, in particular machine learning methods.

A fundamental problem in using machine learning in automated reasoning is a suitable representation of logical formulae. A formula as a solely syntactic object is no longer sufficient and we have to exploit its semantic properties. Various approaches have been proposed for different logical systems. In this paper we will concentrate on the simplest yet very powerful standard logical system—classical (Boolean) propositional logic.

This paper presents, as far as we know, a novel neural representation of propositional formulae that makes it possible to test whether a given formula has a given property, e.g., whether the formula is always true or not. Clearly, we try to solve a well-known coNP-complete problem. However, the fact that the problem is generally hard and requires a non-trivial search does not rule out the possibility that a decent heuristic can be learned, moreover, if only a specific subset of formulae is involved. In particular, our general goal is to obtain a useful heuristic that can help us in guiding a proof search, where we typically face numerous choice points.

Unlike in natural language processing, a parse tree for a formula is available for free. Although some approaches do not exploit this feature and try to learn the structure of a formula on their own, using usually various recurrent neural networks (RNN), it is more common to take advantage of this knowledge. Moreover, it seems that the later approach has a significant edge, see Evans et al. (2018). Usually propositional atoms, the basic building blocks of propositional formulae, are learned as embeddings and each logical connective is treated as a unique neural network that given the vector representation of its arguments produces a vector that represents an application of the connective on these arguments, e.g., a binary connective takes two vectors of length $d$, and produces a new one of length $d$, see Allamanis et al. (2017). This clearly leads to tree recursive neural networks Socher et al. (2012) where the structure of the network follows the parse tree. Such models are built bottom-up and the meaning of the formula is usually the vector produced in the root of the tree.

Our model also uses the parse tree of the formula, but the knowledge is propagated in the opposite direction. We start with a vector (random or learned), representing a property we want to test, and

we propagate it from the root to leaves (propositional atoms). The knowledge propagated to atoms is then processed by recurrent neural networks and a decision is produced. This makes it possible to ignore completely the names of propositional atoms and concentrate more on structural properties of formulae.

The experimental results suggest that the model is more than competitive and beats other known approaches on some benchmarks. More importantly, our model seems to suffer less if bigger formulae are involved and could be more useful in real world scenarios.

The structure of this paper is as follows. In Section 2 we discuss the architecture of our model in full details and also a dataset on which we will experiment is introduced there. In Section 3 we discuss an implementation of building blocks of our network, present experimental data, and shortly describe possible interpretations of our model. Some potential future modifications are briefly mentioned in Section 4. Few relevant models are mentioned in Section 5 and the paper concludes with Section 6.

## 2 MODEL

Our model tries to mimic the following approach sometimes used by humans to decide whether a propositional formula is always true (tautology). If we want to know whether, e.g., $(p \rightarrow q) \vee (q \rightarrow p)$ is a tautology[1] we can suppose the contrary and try to produce a truth-value assignment such that the whole formula is false. It means both $p \rightarrow q$ and $q \rightarrow p$ are false under such an assignment[2]. It follows from the former that $p$ is true and $q$ is false, but from the latter we obtain that $q$ is true and $p$ is false. Because such an assignment is impossible, we showed that the formula cannot be false and hence it is always true.

In a nutshell, we try to propagate a truth value assigned to the whole formula, in our case the formula being false, to its subformulae and we repeat this process until we reach propositional atoms. This gives us a set of constraints on possible truth-value assignments to atoms. At this point we check whether a truth-value assignment satisfying these constraints exists and thus we answer the original question.

However, it is pretty clear that usually this approach is not so straightforward. For example, if we want a formula of the form $A \rightarrow B$ to be true, it means that the subformula $A$ is false or the subformula $B$ is true. Moreover, such choices easily accumulate with the increasing depth[3] of a formula.

Still we will show that a neural model based on the above mentioned idea is competitive. Similarly to the standard recursive approach, we also represent a formula by a recursively built model based on the parse tree of the formula. However, unlike there we use a top-down approach. In our case, we start with a vector $\boldsymbol{w} \in \mathbb{R}^d$ (random or learned), where $d$ is a parameter of the model, that roughly represents a property of formulae we want to check.[4] We continue by propagating this knowledge to subformulae in a top-down manner, e.g., a binary connective $\circ$ takes a vector $\boldsymbol{v} \in \mathbb{R}^d$ (input) and produces two new vectors $\boldsymbol{v}_1 \in \mathbb{R}^d$ and $\boldsymbol{v}_2 \in \mathbb{R}^d$ (outputs), hence $\circ \colon \mathbb{R}^n \rightarrow \mathbb{R}^n \times \mathbb{R}^n$. Since all the vectors have the same length $d$, it is possible to connect individual building blocks together and produce a recursive neural network.

The basic building blocks of the tree part of our model are

$\boldsymbol{w} \in \mathbb{R}^d$      an initial vector,

$c_i \colon \mathbb{R}^d \rightarrow \underbrace{\mathbb{R}^d \times \cdots \times \mathbb{R}^d}_{k\text{-times}}$      a neural network that represents a $k$-ary logical connective $i$,

for all logical connectives $i$ in our language[5].

---

[1] A formula that is true under any assignment of truth values to propositional atoms.

[2] For simplicity, from now on we assume that all the claims are under a given assignment.

[3] The depth of a formula is the depth of its parse tree.

[4] However, it does not mean that by simply changing $\boldsymbol{w}$ our model starts to test a different property, other parts of the model have to be trained accordingly, see also footnote 8.

[5] Note that even nullary connectives, called constants, are allowed.

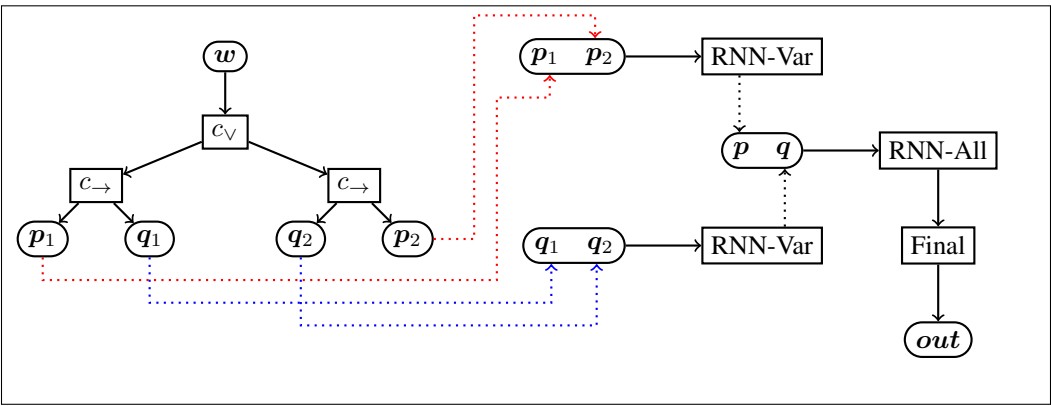

Figure 1: An example of our model for formula $F = (p \to q) \vee (q \to p)$. The initial vector $\boldsymbol{w}$ is propagated through the tree recursive network part producing vectors $\boldsymbol{p}_1, \boldsymbol{p}_2, \boldsymbol{q}_1$, and $\boldsymbol{q}_2$, where $\boldsymbol{p}_1$ corresponds to the first occurrence of the atom $p$ in $F$. Vectors corresponding to the same atom are processed by RNN-Var and these results are then processed by RNN-All. The final output $\boldsymbol{out}$ is produced by Final using the output of RNN-All.

From these components, which are shared among all formulae in a given language, we dynamically build a recursive network for a particular formula that corresponds to the parse tree of the formula—the parse tree tells us how to connect individual components together. For example, in the left part of Figure 1 we see a model for $F = (p \to q) \vee (q \to p)$, which is directly obtained from the parse tree. The vector $\boldsymbol{w}$ is an input to the root of the tree and all the connectives in it are replaced by neural nets $c_i$ that represent them. Atoms contain propagated outputs (vectors) of this tree recursive model, here we obtain vectors $\boldsymbol{p}_1, \boldsymbol{q}_1, \boldsymbol{q}_2, \boldsymbol{p}_2 \in \mathbb{R}^d$, where $\boldsymbol{p}_1$ is at the position of the first occurrence of $p$ in $F$.

Note that each occurrence of the same atom in our tree model produces a unique vector. The fact that they correspond to the same atom is exploited by another level of our model. We take a recurrent neural network RNN-Var that has a sequence of all vectors corresponding to the same atom in the formula as an input and outputs a vector in $\mathbb{R}^d$. In Figure 1 RNN-Var takes $\boldsymbol{p}_1$ and $\boldsymbol{p}_2$ and outputs $\boldsymbol{p} \in \mathbb{R}^d$ and also the same RNN-Var takes $\boldsymbol{q}_1$ and $\boldsymbol{q}_2$ and outputs $\boldsymbol{q} \in \mathbb{R}^d$.

One more recurrent neural network RNN-All takes a sequence of all output vectors produced by RNN-Var and outputs a vector in $\mathbb{R}^d$ that is input to a component Final: $\mathbb{R}^d \to \mathbb{R}^2$ which ends with a softmax layer and produces $\boldsymbol{out} \in \mathbb{R}^2$ that makes it easy to decide whether the output of the whole network is true or false. It should be emphasized that our model completely ignores the names of atoms, we only check whether they are the same or distinct. Moreover, the number of distinct atoms that can occur in a formula is effectively only bounded by the ability of RNN-All to correctly process the outputs of RNN-Var. Hence the model can evaluate formulae that contain more atoms than formulae used for training.

The basic building blocks of the recurrent part of our model are

RNNVar: $\mathbb{R}^d \times \cdots \times \mathbb{R}^d \to \mathbb{R}^d$     a RNN aggregating vectors corresponding to the same atom,

RNNAll: $\mathbb{R}^d \times \cdots \times \mathbb{R}^d \to \mathbb{R}^d$     a RNN aggregating the outputs of RNN-Var components,

Final: $\mathbb{R}^d \to \mathbb{R}^2$     a final decision layer.

It should be emphasized again how the model is built. Given a property we want to test, e.g., whether a formula is always true, we train representations of $\boldsymbol{w}$, $c_i$, RNN-Var, RNN-All, and Final. These components are shared among all the formulae. For a single formula we produce a model (neural network) recursively from them as described above. For implementation details see Section 3.

## 2.1 DATASET

To provide comparable results in Section 3 we use the dataset[6] presented in Evans et al. (2018) and thoroughly described therein. For our purposes here it is essential that it contains triples of the form $(A, B, A \models B)$ where $A$ and $B$ are propositinal formulae and $A \models B$ indicates whether $B$ follows from $A$. The dataset contains train (99876 triples), validation (5000), and various test sets with a different level of difficulty given by a number of atoms and connectives[7] occurring in them.

From the deduction theorem we know that $A \models B$ is equivalent to $A \to B$ being a tautology. We prefer the later form, because it fits nicely into our model and instead of learning the meaning of entailment ($\models$), we can use the connective $\to$ directly.

Note that although our model ignores the names of atoms this cannot be exploited directly, because validation and test sets do not contain pairs of formulae that would result from pairs in the training set by renaming atoms.

## 3 EXPERIMENTS

Although our model is conceptually simple, each building block can be implemented in various ways. In this section we briefly discuss some possible variants and parameters. However, we have not tried to optimize over all the possible parameters discussed later and it is very likely that our results can be improved.

The following implementation of the model introduced in Section 2 is our standard experimental model, called TopDownNet:

- $d = 128$,
- $w \in \mathbb{R}^d$ is a learned vector,
- every $c_i$ is a sequence of linear, ReLU, linear, and ReLU layers, where the input size and output size is always the same with the exception of binary connectives, where the last linear layer is $\mathbb{R}^d \to \mathbb{R}^{2d}$,
- RNN-Var is a gated recurrent unit (GRU) with 2 recurrent layers and the size of the input and hidden state is $d$,
- RNN-All is a GRU with 1 recurrent layer and the size of the input and hidden state is $d$,
- Final is a sequence of linear ($\mathbb{R}^d \to \mathbb{R}^{d/2}$), ReLU, linear ($\mathbb{R}^{d/2} \to \mathbb{R}^2$), and log softmax layers,

and we use the mean square error as a loss function and Adam as an optimizer with the learning rate $10^{-4}$.

A key parameter of our model is the dimension $d$, which is the length of the initial vector $w$ and also the length of many other vectors occurring in our model. Even $d = 8$ produces reasonable results, as Figure 2 shows, but with increasing $d$ the quality of the model grows, see Table 1.

Our various experiments suggest that whether $w$ is chosen randomly or learned makes little to no difference.[8] Similarly and more importantly, connectives $c_i$ composed of a single linear layer seem to perform almost equally, and sometimes even better, to more complicated versions containing non-linearities as in our standard TopDownNet.

In RNN-Var and RNN-All we use gated recurrent units (GRUs) which in our experiments perform similarly to long short-term memory (LSTM) RNNs. Similarly to results obtained in Evans et al. (2018), we have seen no real advantage of using their bidirectional variants for our model and in many cases they produce worse results. The order in which RNN-Var and RNN-All consume their inputs is random.

---

[6]It can be obtained from `https://github.com/deepmind/logical-entailment-dataset`.

[7]The dataset contains negations ($\neg$), implications ($\to$), conjunctions ($\wedge$), and disjunctions ($\vee$).

[8]An obvious question is whether it makes sense to even consider a learned $w$. Generally, it could be useful, because we can be in a situation where we want to use more such vectors which are in relations that we want to learn.

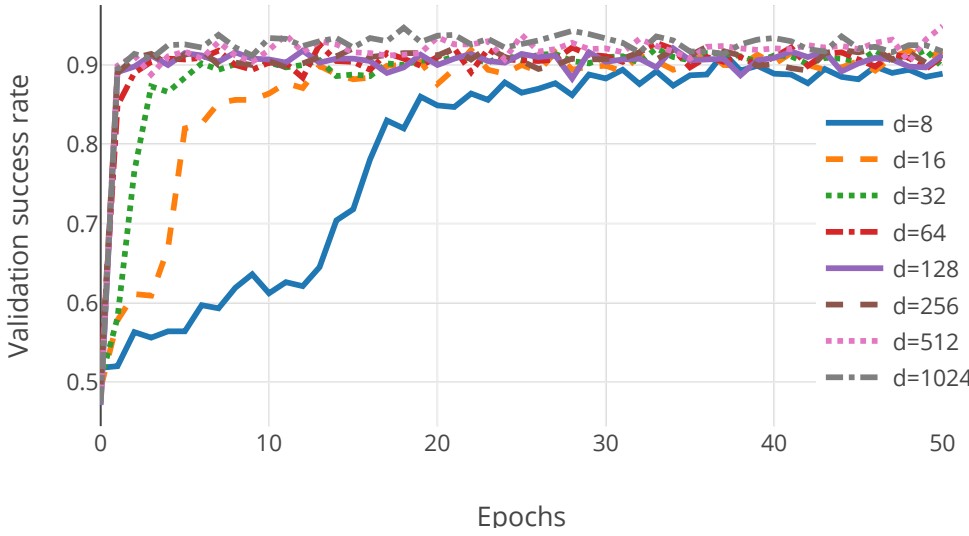

Figure 2: Various choices of $d$ used on a very simple variant of our network with a long short-term memory (LSTM)—every $c_i$ is a linear layer, RNN-Var and RNN-All are LSTMs with one recurrent layer, and Final is a linear layer combined with log softmax.

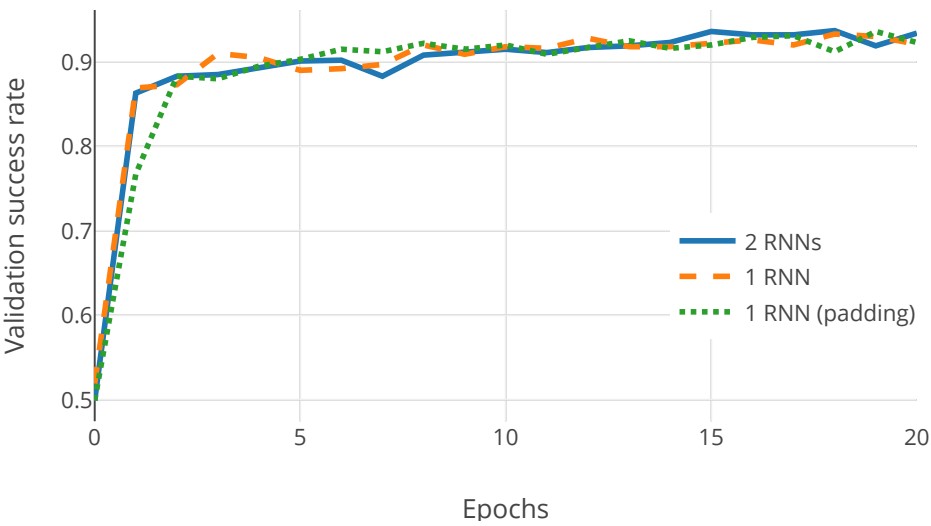

Figure 3: RNN variants where 2 RNNs is TopDownNet ($\boldsymbol{w}$ is random and fixed). 1 RNN uses RNN-Var also as RNN-All and 1 RNN (padding) uses only RNN-Var where the sequences of distinct atoms are separated by padding.

Although it is conceptually nicer and fits to our original intuition, our combination of two RNNs and their functions clearly suggest a question whether they are in fact necessary. Either we can use the same RNN for RNN-Var and RNN-All, or we can use padding to separate the sequences of different atoms. Our preliminary experiments suggest that both new variants and our original approach perform quite similarly, see Figure 3.

It seems that if Final is just a combination of linear and log softmax layers the performance is slighly better than the one used in TopDownNet with an added non-linearity. However, the best performing models that we present in Table 1 use our standard parameters with different values of the dimension $d$.

Table 1: TopDownNet models against other approaches (accuracy)

| model | valid | easy | hard | big | massive | exam |
|---|---|---|---|---|---|---|
| TreeNet Encoders | 72.7 | 72.2 | 69.7 | 67.9 | 56.6 | 85.0 |
| TreeLSTM Encoders | 79.1 | 77.8 | 74.2 | 74.2 | 59.3 | 75.0 |
| PossibleWorldNet | **98.7** | **98.6** | **96.7** | **93.9** | 73.4 | **96.0** |
| | | | | | | |
| TopDownNet ($d = 128$) | 94.0 | 92.8 | 81.0 | 80.7 | 79.7 | 95.0 |
| TopDownNet ($d = 256$) | 95.1 | 95.2 | 82.3 | 80.3 | 82.4 | 95.0 |
| TopDownNet ($d = 512$) | 95.1 | 95.3 | 84.2 | 83.6 | **83.6** | **96.0** |
| TopDownNet ($d = 1024$) | 95.5 | 95.9 | 83.2 | 81.6 | **83.6** | **96.0** |

In Table 1 is a comparison of our models and other approaches mentioned in the benchmark in Evans et al. (2018). The values for the first three models are taken from that paper. TreeNN follows Allamanis et al. (2017) and TreeLSTM follows Tai et al. (2015). PossibleWorldNet is developed in Evans et al. (2018). TopDownNets are our standard models from the beginning of Section 3 with the lowest losses on the validation set for different values of $d$.

The results in Table 1 show that our model is competitive with other approaches. It beats standard tree recursive models (TreeNet Encoders and TreeLSTM Encoders) and PossibleWorldNet, which performs similar to testing random truth-values, on the massive set and keeps pace on examples from textbooks. The good results on the massive set suggest that our model can do better than just random testing of truth-values. Moreover, it is interesting that the increase of $d$ does not have such a significant role, as already mentioned before.

## 3.1 Interpretation of model

We present our model as an alternative to bottom-up approaches, where a possible interpretation of such models is that they combine (samples of) random assignments and produce the value of the whole formula. Similarly, we could argue that our model propagates a truth value top-down and makes (sampled) random choices when multiple options are available, e.g., when we want to make $A \to B$ true. However, this clearly does not explain better results of our model, because for example on the massive set from Table 1 random assignments are much more successful than random top-down propagations.

Our model therefore has to take advantage of something else, or it significantly improves the above mentioned random top-down propagations. Clearly, it uses training data more efficiently, because it is insensitive to the names of atoms, but this is not enough to explain our results. However, it is possible that the top-down propagation captures the overall structure of a formula better and enables a more efficient transfer of knowledge through formulae. Although it is hard to analyze our models completely, we can say something about what is commonly happening at the beginning; close to $\boldsymbol{w}$.

All formulae in the dataset are of the form $A \to B$, see Section 2.1, hence propagating $\boldsymbol{w}$, which is supposed to mean false, through $\to$, denoted $c_\to(\boldsymbol{w})$, produces two vectors $\boldsymbol{v}_{\to 1}$ and $\boldsymbol{v}_{\to 2}$ that correspond to $A$ and $B$, respectively. These two vectors are commonly very distinct and applying negation ($c_\neg$) on one of them almost exactly produces the other one. Moreover, this remains true even if we apply $c_\neg$ repeatedly, see Figure 4. If we now apply $c_\wedge$ on $\boldsymbol{v}_{\to 1}$, assuming $B$ is $C \wedge D$, then we not so surprisingly obtain two vectors very similar to each other and also similar to $\boldsymbol{v}_{\to 1}$; loosely speaking, they should all represent being true. More interesting is to apply $c_\wedge$ on $\boldsymbol{v}_{\to 2}$, assuming $B$ is $E \wedge F$, because we obtain two vectors that are similar, but not as similar as in the previous case and usually even less similar to $\boldsymbol{v}_{\to 2}$. This suggests that something more involved than a simple splitting of truth values according to possible choices can happen in the model.[9] Nevertheless, it

---

[9]Loosely speaking, if we assume that $\boldsymbol{v}_{\to 2}$ is false, then there are three ways how to make the conjunction false $\{(0, 0), (0, 1), (1, 0)\}$, where 0 means false and 1 means true. Hence the first component of the conjunc-

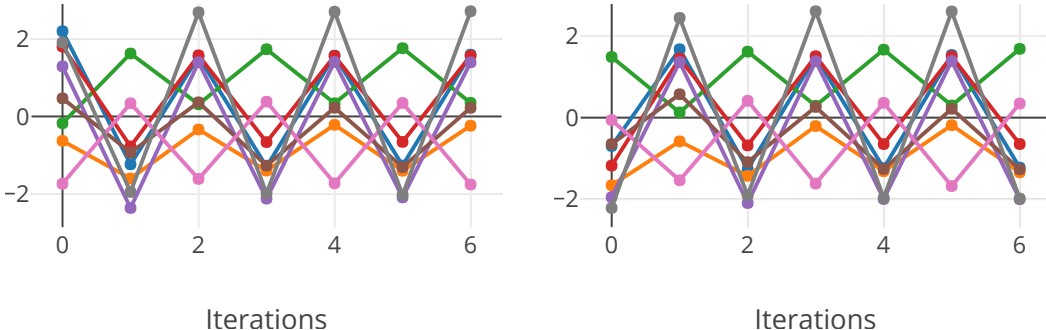

Figure 4: The repeated applications of negation ($c_\neg$) on $v_{\rightarrow 1}$ (left) and $v_{\rightarrow 2}$ (right), which are the results of $c_\rightarrow(\boldsymbol{w})$, for a model where every $c_i$ is a linear layer with $d = 8$ and a learned $\boldsymbol{w}$. The individual components of the vectors are shown.

seems that the model is able to transfer the piece of knowledge that something is (almost) true or false.

Unfortunately, it is very unclear what happens deeper in the model and how recurrent neural networks combine the vectors in the leaves. Moreover, RNN-Var and RNN-All consume their inputs in random order and this slightly influences the results. However, it is plausible, based on initial experiments with small formulae, that RNN-Var, loosely speaking, checks inconsistent assignments to the same atoms and RNN-All just aggregates these results. Moreover, it seems that our model is quite robust to changes in the order of arguments in commutative connectives ($\wedge$ and $\vee$) and to other simple transformations of subformulae—replacing $A \rightarrow B$ by $\neg B \rightarrow \neg A$, $A \wedge B$ by $\neg(\neg A \vee \neg B)$ etc.

A possible explanation of very good performance could be that our model just exploits the given dataset in some unforeseen way. In fact, we want to learn regularities in the dataset. Although this is possible, we believe that this does not explain the performance of our model completely. We trained our model on the BOOL8 and BOOL10 datasets from Allamanis et al. (2017) adapted[10] to our problem and tested it on unseen equivalence classes with the accuracy 98.3% and 80.5%, respectively. These numbers are not directly comparable to the results presented in the paper, however, they suggest that our model is able to perform reasonably even on other datasets. Similarly, if we use the dataset from Evans et al. (2018) to test satisfiability[11], then we obtain results similar to the results on the original dataset.

## 4 POSSIBLE VARIANTS

We aimed for a simple model and hence a plethora of modifications is possible; they can both improve the quality of produced models and their generality. Here we will discuss at least some of them.

Our model uses a feedforward neural network for representing a formula and only then two recurrent networks are used to process the results of the feedforward part. Clearly, more complicated versions can be produced that provide a better interplay between these two layers and/or the feedforward part can become more complicated, e.g., to allow a communication between different subformulae.[12]

---

tion could be represented by $\boldsymbol{v}_1 = (0, 0, 1)$ and the second one by $\boldsymbol{v}_2 = (0, 1, 0)$. If one naturally represents false in $\boldsymbol{v}_{\rightarrow 2}$ by $(0, 0, 0)$, then $\boldsymbol{v}_1$ and $\boldsymbol{v}_2$ should be closer to $\boldsymbol{v}_{\rightarrow 2}$ than to each other.

[10]We produced randomly balanced sets containing pairs of equivalent and non-equivalent formulae.

[11]For every formula $A \rightarrow B$ we take $\neg(A \rightarrow B)$ and propagate negation through the formula. Hence we obtain, e.g., $A \wedge \neg B$ and then continue by propagating negation through $B$.

[12]An example how a more complicated model can help is Peirce's law $((p \rightarrow q) \rightarrow p) \rightarrow p$, a well-known tautology. Similarly as in Section 2 we try to produce a truth-value assignment such that Peirce's law is false. It means that $(p \rightarrow q) \rightarrow p$ is true and $p$ is false. Now using that $p$ is false, we know that $p \rightarrow q$ is also false and hence $p$ is true and $q$ is false, a contradiction thanks to $p$. However, without using the fact that $p$ is false

For example, it is possible to propagate information also back in the bottom-up direction through the tree part of our model, or use an attention mechanism.

It should be also noted that although we use our model for deciding whether a formula is a tautology, we can try to enforce other properties by a similar process, e.g., we can test whether a formula is satisfiable, i.e., is true under some assignment of truth-values. Also learning more related properties, more vectors like $w$, with all the other components shared could be an interesting problem, and their interplay could even improve the performance on individual tasks.

Clearly, our presentation of the model suggests that it heavily relies on properties of classical (Boolean) propositional logic. Nevertheless, it can be used directly also in a non-classical (propositional) setting. Note that the semantics of a given logic is completely learned from examples and no a priory knowledge is provided, however, it is of course possible that our model is unable to reasonably capture the semantics of the given logic and some modifications, e.g., those mentioned above, are necessary. The question whether a more involved model based on our model can be successfully used also for more complex formulae (first-order or even higher-order) is a bit unclear, a clear goal for future research.

## 5 RELATED WORK

A thorough recent comparison of related methods is in Evans et al. (2018) and therefore this section is only very brief. In Wang et al. (2017) authors also develop a model which is invariant to atoms (variables) renaming. However, a formula graph obtained from a parse tree is there translated into a vector using the embeddings of nodes given by their neighborhoods. Moreover, the problem they study is different (first-order and higher-order logics). In Zaremba et al. (2014), a pioneering paper on using recursive neural networks Socher et al. (2012) for mathematical expressions, they also do not deal with variables, because they allow at most one to occur in them.

In Allamanis et al. (2017) a bottom-up approach is presented, however, it also contains a restricted form of backward communication that proved to be useful. The problem studied there is learning vector representations of formulae and based on their similarity test their properties. In our case, we test the properties directly.

An approach based solely on recurrent neural networks has been used in natural language processing where textual entailment is an important topic. For LSTM models with an attention mechanism see Wang & Jiang (2016); Rocktäschel et al. (2016).

PossibleWorldNet developed in Evans et al. (2018) addresses the same problem as our model. It exploits a novel idea of learning various useful assignments and test them latter on. However, the results suggest that the method performs similarly to random assignments (a very powerful technique on its own). It could be because the standard bottom-up approach is used.

## 6 CONCLUSION

We have presented a novel top-down approach to represent formulae by neural networks and showed some preliminary experiments. They suggest that our approach is competitive with other known approaches and beats them on some benchmarks. More importantly, it seems that the presented model can deal better with the increasing size of formulae than other approaches. The model deals only with the structure of formulae and ignores completely for example names of individual atoms, only whether they are the same or distinct matters.

ACKNOWLEDGMENTS

This work was supported by the European Regional Development Fund under the project AI&Reasoning (reg. no. CZ.02.1.01/0.0/0.0/15_003/0000466).

---

(inaccessible to our model) in processing $(p \rightarrow q) \rightarrow p$ we have to deal with more choices along this line of reasoning.

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
