# OpenReview forum: "Top-Down Neural Model For Formulae"
_ICLR.cc/2019/Conference_

### Official Review · AnonReviewer1 · 2018-10-31

**Rating:** 6
**Confidence:** 2

**Review:**

In this paper the authors propose a neural model that, given a logical formula as input, predicts whether the formula is a tautology or not. Showing that a formula is a tautology is important because if we can classify a formula A -> B as a tautology then we can say that B is a logical consequence of A. The structure of the formula is a feedforward neural network built in a top-down manner. The leaves of this network are vectors (each of them represents a particular occurrence of an atom) which, after the construction of the formula, are processed by some recurrent neural networks.

The proposed approach seems interesting. However, my main doubt concerns the model. It seems to outperform the state-of-the-art, but the authors do not give any explanations why. There is no theoretical or intuitive explanation of why the model works. Why we need RNNs and not feedforward NNs? I think this is an big issue.
In conclusion, I think that the paper is a bit borderline. The model should be better explained. However, I think that the approach is compelling and, after a minor revision, the paper could be considered for acceptance.

[Minor comments]
Page 4.
“The dataset contains train (99876 pairs)”, pairs of what?

Page 5.
What is the measure of the values reported in Table 1? Precision?

---

> ### Author Response · Authors · 2018-11-27
> **Reply to your review**
>
> Thank you for your comments. You are right that the inner working of the model is unclear. A brief Section 3.1 was added to a revised version of the paper, where the produced models are shortly analyzed. Hopefully, it sheds some light on the model.
>
> Concerning your second point, RNNs are used because they fit nicely into the model. It makes it possible to have potentially unlimited number of occurrences of an atom and the number of distinct atoms in a formula, which is a nice feature of the model. Hence the model can evaluate formulae that contain more atoms than formulae used for training. It is possible to use feedforward NNs, but it seems that we then mimic the unfolding of RNNs.
>
> Both your minor comments were incorporated into a revised version of the paper.

---

> ### Comment · Area_Chair1 · 2018-11-30
> **Please clarify position**
>
> Thank you, reviewer 1, for your review. I appreciate and understand your position regarding the lack of explanation for the model's performance. However, our field is primarily empirical, and it is common for engineering-oriented papers to produce such results which will only be properly understood and explained by further work. The literature is rife with examples, from GANs to regularization tricks for RNNs. You must ask yourself: are the results sufficiently believable? is the study conducted rigorously? and have the authors attempted to explain and discuss them to a reasonable extent? Please read the author response, revisions to the paper, and be prepared to reconsider your assessment or provide further justification as to why you stand by your current score, if that is what you choose to do.

---

> > ### Comment · AnonReviewer1 · 2018-12-04
> > **Updated Review**
> >
> > I've read the new version of the paper and the comments of other reviewers and I've decided to increase my score.

---

### Official Review · AnonReviewer3 · 2018-10-31
**Good but under-explored performance of a semi-original approach to an important problem in neural-symbolic computing**

**Rating:** 6
**Confidence:** 4

**Review:**

Cons

1.	There is no study of the representations developed by the model, which is unfortunate because this is a conference on learning representations and because there is little light shed on how the network achieves its rather high level of performance.
2.	It seems less generally useful to have such a special-purpose network for computing global properties like tautologicality than to have a network that produces actual vector encodings of propositions, as typical of the bottom-up tree-structured models.

Pros

3.	The paper is quite clear.
4.	The problem is important.
5.	The paper pursues the familiar path of a tree-structured network isomorphic to the parse tree of a propositional-calculus formula, but with the original twist of passing information top-down rather than bottom-up.
6.	The results are impressively strong. In particular, it improves by 10% absolute over the special-purpose and highly performant PossibleWorldNet on the most difficult category of problems, the ‘massive’ category, achieving 83.6% accuracy.

Pro/Con mix

7.	Although the paper did not provide much insight into what was going on in the network to allow it to perform well (point 1 in ‘Cons’), I was able to convince myself I could understand a way the architecture *could* succeed (whether this possible approach matches the actual processing in the model I have no way of assessing). In brief, the vector that is passed down the network can be thought of as a list of truth values across multiple possible worlds of the tree node at which the vector resides. To search for a counterexample to tautologicalhood, the original input vector to the root node could be the zero (false) vector. If the kth value in the vector at a parent node labeled ‘or’ is 0 (the disjunction is false in world k) then in the two children the kth value must also be 0. If the kth value of the vector at an XOR node is 0, the kth value of the two children must both be 0 or both be 1; actually these values need not reside in position k so the children could both have value 0 at some position i and both have value 1 at another position j. Then in the RNN-Var component of the network, which checks for consistency across multiple tokens of the same proposition variable, each position k in all vectors for the same variable can be checked for equality, producing a value 1 in the output vector if all have value 1, producing 0 if all have value 0, and producing value -1 if the values do not all agree. Then RNN-All checks across all vectors for proposition variable types to see if there’s a position k in which no value -1 occurs; if so, the values of the variable vectors at position k give the truth values for all variables such that the overall proposition has the desired value 0: a counterexample exists. If no such position k exists, the proposition is a tautology. This seems roughly right, at least.

---

> ### Author Response · Authors · 2018-11-27
> **Reply to your review**
>
> Thank you for your comments. A new Section 3.1 was added to a revised version of the paper, where the inner working of the model is briefly discussed. Although it is definitely far from being conclusive, it, hopefully, sheds some light on the model.
>
> Your description (point 7) of how the model can possible work corresponds to the idea behind the model as described in Section 2 and discussed in new Section 3.1. An interesting point in your text is that values may change their positions in lists of truth values. In fact, something like that can actually happen, but so far, it is really unclear how to do this, because such changes have to be (almost) consistent through the whole model. Moreover, to make things even more complicated, different atoms occur at different levels (their depth) in a formula.
>
> You are right (point 2) that the model, in its current form, cannot produce suitable vector encodings of propositions. For example, the model is invariant to the renaming of atoms. However, for formulae where this is no longer an issue, e.g., sentences in FOL, it is possible to imagine such interpretations even using a top-down approach.

---

### Official Review · AnonReviewer2 · 2018-11-03
**Simple interesting neural-net model of logical formulae**

**Rating:** 6
**Confidence:** 3

**Review:**

In this paper, the authors provide a new neural-net model of logical formulae. The key feature of the model is that it gathers information about a given formula by traversing its parse tree top-down. One neural net of the model traverses the parse tree of the formula from the root all the down toward the leaves, and generates vectors for the leaves of the tree. Then, another RNN-based neural net collects these generated vectors, and answers a query asked for the formula, such as logical entailment. When experimented with Evans et al.'s data set for logical entailment queries, the authors' model outperforms existing models that encode formulae by traversing their parse trees bottom-up.

I found the idea of traversing a parse tree of a formula top-down and converting it to a vector very interesting. It is also good to know that the idea leads to a competitive model for at least one dataset.

However, I am hesitant to be a strong supporter for this paper. I feel that the cons and pros of the model and its design decisions are not fully analyzed or explained in the paper; when reading this paper, I wanted to learn a rule of thumb for deciding when (and why if so) a top-down model of logical formulae works better than a bottom-up model. I understand that what I ask for is very difficult to answer, but experiments with more datasets and different types of queries (such as satisfiability) might have made me happier.

Here are some minor comments.

* Abstract: I couldn't quite understand your point about atoms. According to Figure 1, there is a neural net for each propositional symbol, and this means that your model tracks information about which occurrences of propositional symbols are about the same one. Is your point about the insensitivity of your model to a specific name given to each symbol?

* p1: this future ===> this feature

* p2: these constrains ===> these constraints

* p2: recursively build model ===> recursively built model

* p2: Change the font of R in the codomain of ci.

* p3: p1 at the position of ===> p1 is at the position of

---

> ### Author Response · Authors · 2018-11-27
> **Reply to your review**
>
> Thank you for your comments. Indeed, the question why and when a top-down model outperforms a bottom-up model is crucial. However, as you have pointed out, it is likely a difficult question to answer. A new Section 3.1 was added to a revised version of the paper, where the inner working of the model is briefly analyzed. A top-down model was also tested on formulae from another dataset. Although the results are hard to compare directly, it seems that the model does not exploit just one particular dataset. Similarly, we can reformulate a TAUT-problem as a SAT-problem by taking the negation of formula. The results remain similar on the dataset from Evans et al., however, this is hardly surprising, because the problem remains essentially the same from the point of view of a top-down approach.
>
> All your minor comments were incorporated into a revised version of the paper.

---

### Meta-Review · Area_Chair1 · 2018-12-13
**Probably acceptable**

**Confidence:** 3
**Recommendation:** Accept (Poster)

**Metareview:**

This paper presents a method for building representations of logical formulae not by propagating information upwards from leaves to root and making decisions (e.g. as to whether one formula entails another) based on the root representation, but rather by propagating information down from root to leaves.

It is a somewhat curious approach, and it is interesting to see that it works so well, especially on the "massive" train/test split of Evans et al. (2018). This paper certainly piques my interest, and I was disappointed to see a complete absence of discussion from reviewers during the rebuttal period despite author responses. The reviewer scores are all middle-of-the-road scores lightly leaning towards accepting, so the paper is rather borderline. It would have been most helpful to hear what the reviewers thought of the rebuttal and revisions made to the paper.

Having read through the paper myself, and through the reviews and rebuttal, I am hesitantly casting an extra vote in favour of acceptance: the sort of work discussed in this paper is important and under-represented in the conference, and the results are convincing. I however, share the concerns outlined by the reviewers in their first (and only) set of comments, and invite the authors to take particular heed of the points made by AnonReviewer3, although all make excellent points. There needs to be some further analysis and explanation of these results. If not in this paper, then at least in follow up work. For now, I will recommend with medium confidence that the paper be accepted.